# Epigenetics and Heart Failure

**DOI:** 10.3390/ijms21239010

**Published:** 2020-11-27

**Authors:** Syeda Shegufta Ameer, Mohammad Bakhtiar Hossain, Ralph Knöll

**Affiliations:** 1ICMC (Integrated Cardio Metabolic Centre), Myocardial Genetics, Karolinska Institutet, Novum, Hiss A, Våning 7, Hälsovägen 7-9, 141 57 Huddinge, Sweden; Shegufta.Ameer@ki.se (S.S.A.); Ralph.knoell@ki.se (R.K.); 2Bioscience Cardiovascular, Research and Early Development, Cardiovascular, Renal and Metabolism (CVRM), BioPharmaceuticals R&D, AstraZeneca, 431 50 Gothenburg, Sweden

**Keywords:** epigenetics, DNA methylation, heart failure

## Abstract

Epigenetics refers to changes in phenotypes without changes in genotypes. These changes take place in a number of ways, including via genomic DNA methylation, DNA interacting proteins, and microRNAs. The epigenome is the second dimension of the genome and it contains key information that is specific to every type of cell. Epigenetics is essential for many fundamental processes in biology, but its importance in the development and progression of heart failure, which is one of the major causes of morbidity and mortality worldwide, remains unclear. Our understanding of the underlying molecular mechanisms is incomplete. While epigenetics is one of the most innovative research areas in modern biology and medicine, compounds that directly target the epigenome, such as epidrugs, have not been well translated into therapies. This paper focuses on epigenetics in terms of genomic DNA methylation, such as 5-methylcytosine (5mC) and 5-hydroxymethylcytosine (5hmC) modifications. These appear to be more dynamic than previously anticipated and may underlie a wide variety of conditions, including heart failure. We also outline possible new strategies for the development of novel therapies.

## 1. Introduction

Cardiovascular diseases (CVD) have continued to rise over the last few years and they are now considered a modern-day global epidemic, as they affect 1% to 2% of the total population. Heart failure, an important aspect of CVD, encompasses a wide range of conditions, extending from myocardial infarction to congenital heart disease and various cardiomyopathies. Most of these are heritable [1], which means they are due to gene mutations [2] or multiple rare variants [3]. While genetics has significantly contributed to our understanding of different human diseases, the effects of epigenetics on the development and progression on these diseases are not so well understood. It is widely accepted that an interplay between inherited traits and environment plays a major role in the development of cardiac diseases (Figure 1).

Epigenetics refers to changes in phenotypes without changes in genotypes, a concept first described by Waddington in 1942 [4]. Subsequent research has shown that epigenetic modifications can occur as a result of at least three different mechanisms: (i) genomic DNA methylation, (ii) modification of DNA interacting proteins, and (iii) microRNAs. Various types of genomic DNA (gDNA) methylation are known. The most important one is probably methylation of cytosine in the 5′ position. This creates 5-Methylcytosine (5mC) via de novo methyltransferases (DNMTs), and the 5mC is then converted to 5hmC via ten-eleven translocation methylcytosine dioxygenase (TET) enzymes, where the C is followed by a guanine (G) and separated by a phosphate (CpG).

Higher levels of DNA methylation lead to chromatin condensation, for example, as observed during X-chromosome inactivation transposon silencing and genomic imprinting [5]. Until recently, 5mC was thought to be a stable modification of DNA and was expected to remain unchanged throughout the life of a fully differentiated somatic cell. However, it has now been discovered that 5mC can be oxidized to 5hmC [6]. Recent studies have shown that the enrichment of 5hmC on the gene body has been associated with activation of transcription [6,7,8,9,10,11] and TET-mediated 5mC oxidation regulates the activity on the transcription start site (TSS) [7,12,13] and active enhancers. Epigenetic studies of human hearts have reported altered 5mC in chronic heart failure. However, our knowledge with regard to the underlying molecular mechanisms in CVD remains incomplete, despite the observation that DNA methylation appears to regulate CVD development. 

The application of cutting-edge technology is the key to accelerating our understanding of biological mechanisms in health and diseases. This is particularly true for our understanding of the epigenetic mechanisms that control gene expression (GE) in cardiac tissue and the potential implications for heart failure. That knowledge will ultimately drive forward the development of new therapeutic targets and personalized medicine.

## 2. Transcription Factors and Chromatin Crosstalk in Experimental Heart Failure

Genetic differences between inbred mouse strains have been shown to contribute to various cardiac phenotypes [14]. A cardiovascular epigenomics study conducted by Chen et al. in 2016 showed that DNA methylation differed in isoproterenol-induced heart failure between susceptible BALB/cJ and resistant BUB/BnJ mouse strains [15]. Although the group of transcription factors that was regulated by methylation was similar in both of the mouse cohorts, both strains showed different preferences for transcription factors whose target genes were most enriched for altered DNA methylation. Moreover, altered basal DNA methylation leads to a global shift in chromatin accessibility and seems to coordinate with histone-based chromatin marks and poised susceptibility to heart failure before exposure to stress. 

A significant study by Greco et al. described the dynamics of 5hmC in embryonic, neonatal, adult, and hypertrophic mouse cardiomyocytes. They reported that pathological hypertrophy was characterized by a shift toward a neonatal 5hmC distribution pattern [16]. Neonatal cardiomyocytes are enriched in 5hmC on the gene body and these become demethylated during development. This is also an epigenetic mark of the methylome of the adult heart and similar to what has been reported in the human brain [17]. Greco et al. reported that gene ontology analysis of hypertrophied cardiomyocytes displayed the loss of 5hmC within the genic regions of the enzymes of the TCA cycle, namely, mitochondrial fatty acid oxidation and calcium handling [18,19]. The presence of high levels of 5hmC has also been shown to be co-expressed with activating histone marks, such as H3K79me2, H3K9ac, H3K27ac, and H3K4me3. In these cases, 5hmC could be a pre-activating mark that precludes the binding of chromatin modelling proteins. Research has shown that 5hmC depletion significantly affects the enrichment of the activating histone mark H3K27ac within the enhancer region. Moreover, the authors reported that the accumulation of 5hmC was positively correlated with GE and, in particular, with cardiac-specific genes. This results in the re-activation of fetal genes, such as *Myh7* [16].

Another series of experiments was carried out by Rosa-Garrido et al. The authors performed genome-wide chromatin conformation capture (Hi-C) and DNA sequencing on adult cardiac myocytes following transverse aortic constriction (TAC)-induced hypertrophy. Cardiac-specific deletion of CTCF, which is a transcriptional regulator protein with 11 highly conserved zinc finger domains, was generated in mice, which then went on to develop heart failure. Interestingly, human patients with heart failure, who were receiving mechanical unloading via left ventricular assist devices, showed an increased abundance of CTCF. The authors reported that heart failure involved decreased stability of chromatin interactions around the genes that caused the disease. Moreover, pressure overload or CTCF depletion remodeled long-range interactions of cardiac enhancers and these finally resulted in a significant decrease in local chromatin interactions around these functional elements. In conclusion, epigenetic modification has been shown to control both genomic structure and GE in a cell type-specific manner [20,21]. 

Dilated cardiomyopathy (DCM) is a severe form of heart failure, which is characterized by enlarged cavities and decreased systolic function [22]. It can also be described as heart failure with reduced ejection fraction (HFrEF). Another common form of heart failure has been associated with hypertrophic cardiomyopathy (HCM) or left ventricular hypertrophy (LVH). These forms of heart failure are characterized by thickening of the myocardium or changes to LV wall thickness and internal diameter. In such cases, cardiac output is limited by impaired filling and outflow, which is probably a type of heart failure with preserved ejection fraction (HFpEF). Up to 50% of all DCM cases are genetic and involve mutations in titin (TTN), sarcomeric, cytoskeletal, and Z-band genes [23]. Cysteine rich protein 3 (Csrp3, also known as muscle LIM protein (Mlp)) localizes to the sarcomeric Z disc, where it interacts with TTN via the titin capping protein (Tcap, also known as telethonin). Mutations in Csrp3 can lead to myocardial hypertrophy, followed by DCM and heart failure [24]. In 2002, it was suggested that colocalization and interaction of Csrp3 and Tcap are necessary for sarcomeric function. This means that human CSRP3 mutations, for example, CSRP:p.W4R, are associated with DCM [25] or HCM [26]. Another study on Csrp3^+/−^ mice by Heineke et al. in 2005 showed the importance of calcineurin, a Ca^2+^/calmodulin dependent protein phosphatase, in cardiomyocyte function [27]. Calcineurin dephosphorylates and translocates NFAT (nuclear factor of activated T cells) to the nucleus, thereby affecting downstream GE. Furthermore, a trimeric complex is formed at the Z disk, along with α-actinin and calsarcin-1 [28], indicating that Csrp3 is required for calcineurin-NFAT activation and calcineurin anchorage to the Z disk [29]. 

The reversible acetylation of lysine residues within a protein is considered a biologically relevant modification, which is comparable to phosphorylation or other types of post-translational modifications [30]. This acetylation machinery comprises three distinct components. The first is histone acetyltransferases (HATs), or “writers”, which are enzymes that transfer acetyl groups to histone tails within chromatin, generally relaxing the nucleosomal structure and increasing the accessibility of the genes to DNA binding elements. In contrast, the second component, histone deacetylases (HDACs), “erases” the acetylation marks, condenses chromatin, and decreases its local accessibility. The third component is the BET family of bromodomain (BRD) proteins, which functions as “readers” of those acetylation marks and facilitates the protein complex formation required for appropriate gene regulation. Together, this elegantly orchestrated three-component system is a highly dynamic and effective means of governing gene transcription, as shown in a recent review [31]. 

Similarly, it has also been shown that CSRP3 is a target of HATs, such as PCAF, and of HDAC4, as Csrp3 null mice do not show increased calcium sensitivity when treated with HDAC inhibitors (e.g., trichostatin-A). These also establish the regulatory role of acetylation by HATs and HDACs in muscle contraction [32]. Furthermore, heme oxygenase 1 (HO-1) regulates the nuclear accumulation of CSRP3 and plays a major role in myocyte contractility and cardiac function [33].

Although CSRP3 is part of a mechanosensory signalosome, and its DCM and HCM mutations are a known cause of both, the underlying molecular mechanisms are still not well understood, especially with regard to epigenetics [34]. However, a link between epigenetic regulation, calcium sensitivity, and contractility is now evident, but it needs to be clarified further. This will indicate to what extent this link can be utilized for novel drug development. 

## 3. Toward Epigenetic Hallmarks in Human Heart Failure 

The heart shows distinct features during its physiological growth from fetal to adult stages. These include changes from anaerobic to aerobic metabolism, known as the switch to beta-oxidation. They also include the changes in isoform expression of sarcomeric components, including the switch from alpha to beta myosin heavy chains and changes in troponin C isoform. This transcriptional profile appears to be orchestrated by master transcription factors and epigenetic modifications, which ultimately allow these molecular adaptations. 

It has been well documented that fatty acid utilization decreases during cardiac hypertrophy and heart failure, while glycolysis increases [35,36,37,38]. In addition, heart failure has been associated with the fetal pattern of GE. In rats with spontaneously hypertensive heart failure (SHHF), the mRNA levels of mitochondrial fatty acid oxidation enzymes were downregulated by more than 70% during both LVH and heart failure, compared to controls without SHHF [39]. Furthermore, failing human hearts showed a loss of MYH6, the alpha myosin heavy chain, and an increased ratio between MYH7, the beta myosin heavy chain, and MYH6, which may lead to the downregulation of fatty acid oxidation [40,41]. Interestingly, these changes reverse the postnatal switch from anaerobic to aerobic metabolism. It also significantly interferes with the regenerative potential of the myocardium, by inducing cardiomyocyte cell-cycle arrest through the DNA damage response [42]. The importance of epigenetic molecular mechanisms, including 5mC, are well established during embryonic development, but there has been less evidence with regard to 5hmC. However, the role of these modifications in heart failure has not been well defined.

Gilsbach et al. reported that DNA methylation was highly dynamic during cardiomyocyte development, postnatal maturation, and disease. The authors analyzed cardiac tissues and isolated cardiomyocytes after DNA bisulfite treatment during prenatal development. They also examined the postnatal maturation of the heart from infancy to adulthood and during terminal failure. The 5mC level was largely demethylated in cardiomyocyte-specific genes that encoded the sarcoplasmic reticulum Ca^2+^ APTase SARCA2A (*Atp2a2*), cardiac ryanodine receptor (*Ryr2*), the α_1C_-subunit of the L-type Ca^2+^ channel (*Cacna1c*), and titin (*Ttn*). Other genes include mitochondrial components, skeletal troponin I (*Tnni1*), and cardiac troponin I (*Tnni3*), which have been associated with altered calcium sensitivity of the sarcomere. The authors observed 440 genes with epigenetic changes and altered GE in the mouse and human samples and found that they occurred continuously from the fetal stage to adulthood. Moreover, they observed that actively expressed genes had H3K4me1, H3K27ac, and H3K4me3 but lacked H3K27me3. In contrast, the gene bodies of inactive genes showed abundant H3K27me3, with high levels of CpG methylation [43]. These findings were confirmed by Gilsbach et al., who reported that a dynamic interplay between mCpG and histone modification shaped cardiomyocyte transcriptome during development and maturation, and this included human hearts. The investigators only observed six genes with differential methylation in adult failing cardiomyocytes that did not display consistent changes in GE [44]. This data echoed, to some extent, reports that genome-wide chromatin compartments showed no, or only subtle, changes in mouse cardiac myocytes during transverse aortic constriction [20,21], as discussed above.

Although we focused on the role of CpG methylation in cardiac function, histone modifications play an equally important role, particularly in increased afterload conditions. H3K27ac and H3K36me can explain up to 50% of pathological GE and can serve as predictive markers for failing cardiomyocytes, even without accompanied alterations in CpG methylation [44].

In contrast to heart failure induced by TAC, one study showed that coronary occlusions induced a completely different GE program and chromatin accessibility algorithm [45]. Border zone cardiac myocytes lost accessibility for regulatory elements containing the transcription factor MEF2, while injury-associated enhancers were more accessible for AP-1 binding sites. Therefore, different stimuli, such as increased afterload, and possibly pre-load, or ischemia, could induce separate changes in the transcriptome, which are very likely to have been orchestrated via different epigenetic mechanisms. Furthermore, Movassagh et al. used a MeDIP approach, followed by bisulfite sequencing, which identified three angiogenesis-related genetic loci in human end-stage heart failure [46]. Another recent report described the dynamics of 5hmC in embryonic, neonatal, adult, and hypertrophic mouse cardiomyocytes. The authors found that pathological hypertrophy was characterized by a shift toward a neonatal 5hmC distribution pattern [16], as discussed above.

The switch from *Myh6* to *Myh7* expression is an important feature of specific sets of GE related to heart failure per se [41]. Increasing the *Myh7/Myh6* ratio by overexpression of *Myh7* has been shown to cause heart failure [47]. Expression of the fetal cardiac α-myosin heavy chain (MHC) gene *Myh7* has been highly linked to both genic- and enhancer-associated hydroxymethylation, specifically in hypertrophic cardiomyocytes. TET2 was one of the most abundantly expressed enzymes in cardiomyocytes and it appears to regulate GE in a very specific manner. This is evident from the fact that the downregulation of *Tet2* does not affect global hydroxymethylation, but only at specific 5hmC loci [16]. One of these specific 5hmC accumulations was associated with the marked downregulation of *Myh7* gene expression. 

Unfortunately, no epigenetic correlation in terms of 5mC and 5hmC at single base pair resolution has been described. Several different mechanisms have been proposed to explain the alpha to beta myosin heavy chain switch, including calcineurin/NFAT signaling [48], miRs, such as miR-208 [49,50], and thyroid metabolism [51]. Despite this, alterations in 5mC and 5hmC cannot be excluded. Meder et al. generated a genome-wide DNA methylation profile in DCM patients and identified 59 differentially methylated CpG loci in DCM, when they were compared to clinical controls. Of these, 29 were hypomethylated and 30 were hypermethylated. Moreover, the authors were able to link 517 epigenetic loci with DCM and GE. Remarkably, this study reported the overlap of methylation patterns between myocardium and peripheral blood, which included demethylation of NPPA natriuretic peptide A or ANP (NPPA) and natriuretic peptide B or BNP (NPPB) loci [52]. A recent study also identified five unique differentially methylated regions in hypertrophic obstructive cardiomyopathy (HOCM). These included 151 in DCM and 55 in ischemic cardiomyopathy, together with a total of 209 genes associated with these regions [53].

## 4. Conclusions and Outlook

The combined findings of the studies that have been published to date have demonstrated that the alternation of DNA methylation has been associated with experimental types of heart failure or can be identified in human cardiomyopathies. However, different etiologies of heart failure may generate different DNA methylation patterns, and the authors could not assign specific epigenetic modifications for different cardiac disease models such as HCM and DCM. Future work in this field will likely identify these specific mechanisms and lay the foundation for novel drug discovery.

Next generation sequencing, combined with bisulfite and oxidative bisulfite sequencing and single cell technologies, have provided a wealth of knowledge with regard to genomic DNA methylation during cardiac development and disease. These epigenetic modifications play essential roles in processes such as cardiac differentiation, development, postnatal growth, and disease in experimental animal models and during conditions leading to, or underlying, heart failure in patients. The cardiac epigenome is shaped during heart development and remains highly dynamic, both in atrial and ventricular cardiomyocytes. This also seems to be the case for other cardiac cells, despite a considerable scarcity of data. However, the dynamicity appears dependent on the condition or disease, as changes in pressure overload appear less pronounced when they are compared to those that occur during ischemic events.

The overwhelming majority of gDNA methylation data relates to 5mC, which means that complete 5mC and 5hmC genomic data sets are needed. Ideally, these should be available at single-cell resolution and be put into the context of other omics data, including the genome, proteome, metabolome, miRNAome, and overall GE (Figure 1). Moreover, other cell types need to be considered, including fibroblasts, smooth muscle cells, and endothelial cells. Last but not least, left versus right ventricular tissue needs to be analyzed, as well as atrial versus ventricle and mitochondrial epigenetics.

While our knowledge of the epigenome has significantly increased over the last few years, it remains limited and incomplete, and this is particularly true when it comes to epigenetic changes during heart failure. This lack of knowledge relates to the epigenetic changes caused by various experimental strategies that aimed to increase preload or afterload conditions, experimental or pathological ischemia, or changes induced by mutations in cardiomyopathy candidate genes or in the context of different types of human heart failure. Therefore, more research is needed to investigate epigenetic effects under in vitro and in vivo conditions, as well as in various experimental conditions. In addition, a greater focus is necessary on the development and use of epidrugs. Apabetalone, a bromodomain (BRDU) inhibitor, has recently been tested in a large phase III clinical trial and the results only narrowly missed statistical significance [54] (updated findings were disclosed in AHA-2019). However, it might be possible to improve this compound further, to increase efficiency and mitigate the associated safety concerns. Further studies are expected to unravel the role of epigenetics in heart failure in more detail. 

## Figures and Tables

**Figure 1 ijms-21-09010-f001:**
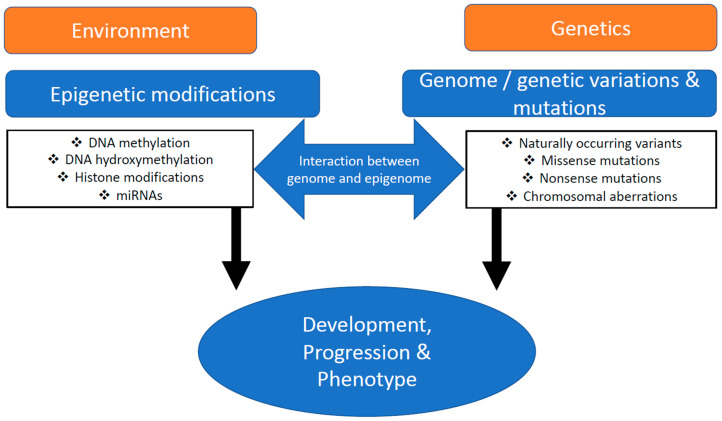
The interplay between inherited traits and environment. A combination of genetic markers and epigenetic effects determine the phenotype.

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
