# Peer review of "Epigenetics and Heart Failure"

_ijms, 2020, doi:10.3390/ijms21239010_

Round 1

Reviewer 1 Report

This is a well-written review on the relationship between epigenetic alterations and cardiac phenotypes. I have no major revisions, although I was curious about whether epigenetic alterations across studies were specific to the genes mentioned, or if the epigenomes of cardiac tissues were widely dysregulated. Similarly, did these patients/animals exhibit epigenetic aberrations in other tissues than heart? This could be something to include as a discussion point if the authors wish.

minor comments:

page 3, line 30: "controls" should be "control" (grammar)

Author Response

This is a well-written review on the relationship between epigenetic alterations and cardiac phenotypes. I have no major revisions, although I was curious about whether epigenetic alterations across studies were specific to the genes mentioned, or if the epigenomes of cardiac tissues were widely dysregulated. Similarly, did these patients/animals exhibit epigenetic aberrations in other tissues than heart? This could be something to include as a discussion point if the authors wish.

Response: The authors thank the reviewer for very relevant comments. The available literature does not allow the authors to comment on whether any specific gene is common across species behind cardiac phenotype. Similarly, most of the studies looking into epigenetic alterations and cardiac phenotype focused on cardiac tissues.

minor comments:

page 3, line 30: "controls" should be "control" (grammar)

Response: Modified, as suggested.

Reviewer 2 Report

The authors have done a comprehensive review of the association between epigenetics and heart failure. They reviewed important original experimental papers and proposed novel therapeutic targets. This paper is well written and provide important information to the readers.

You presented many experimental studies, but many readers may not be able to understand well these data. Then, please make a summary table of the 2. Transcription factors and chromatin crosstalk in experimental heart failure, and 3. Toward epigenetic hallmarks in human heart failure. It is necessary to organize epigenetic mechanisms for each disease model such as HCM, DCM, and ischemic heart disease.

If possible, please provide details on the effects of BRDU inhibitors on human heart failure referred to in No. 54. Additionally, if you have any other human studies data, please present them.

Please provide limitation and future perspective in 4. Conclusion and outlook.

Author Response

The authors have done a comprehensive review of the association between epigenetics and heart failure. They reviewed important original experimental papers and proposed novel therapeutic targets. This paper is well written and provide important information to the readers.

Response: Thank you.

You presented many experimental studies, but many readers may not be able to understand well these data. Then, please make a summary table of the 2. Transcription factors and chromatin crosstalk in experimental heart failure, and 3. Toward epigenetic hallmarks in human heart failure. It is necessary to organize epigenetic mechanisms for each disease model such as HCM, DCM, and ischemic heart disease.

Response: The authors feel that these are very important and relevant comments. However, McNally and Mestroni recently published a comprehensive review on the crosstalk between different transcription factors and chromatin in experimental and human heart failure [1], and this subject is beyond the scope of this current review. The authors also feel that the currently available knowledge / literature does not allow to identify distinct epigenetic mechanisms for different cardiac disease models like, HCM and DCM. These factors are mentioned as limitations of the current work (Page 5, line 51).

If possible, please provide details on the effects of BRDU inhibitors on human heart failure referred to in No. 54. Additionally, if you have any other human studies data, please present them.

Response: The authors would like to provide further details of the BRDU inhibitor study in human heart failure, but they could not find any updated report which was peer reviewed and published.

Please provide limitation and future perspective in 4. Conclusion and outlook.

Response: Thank you for the comment. Please see Page 5, line 51.

Reference:

  1. McNally, E.M.; Mestroni, L. Dilated Cardiomyopathy: Genetic Determinants and Mechanisms. Circ Res 2017, 121, 731-748, doi:10.1161/CIRCRESAHA.116.309396.